# Is the Survivability of Silver Fir under Condition of Strong Ungulate Pressure Related to Mycobiota of Bark-Stripping Wounds?

**Wojciech Pusz** [1], **Anna Baturo-Cieśniewska** [2], **Agata Kaczmarek-Pieńczewska** [1,*], **Katarzyna Patejuk** [1] and **Paweł Czarnota** [3,4]

1  Department of Plant Protection, Division of Plant Pathology and Mycology, Wrocław University of Environmental and Life Sciences, Plac Grunwaldzki 24a, 50-363 Wroclaw, Poland; wojciech.pusz@upwr.edu.pl (W.P.); katarzyna.patejuk@upwr.edu.pl (K.P.)

2  Laboratory of Molecular Mycology, Phytopathology and Entomology, Department of Biology and Plant Protection, UTP University of Science and Technology, Aleja Kaliskiego 7, 85-796 Bydgoszcz, Poland; baturo-a@utp.edu.pl

3  Department of Ecology and Environmental Protection, University of Rzeszów, Zelwerowicza 4 Street, 35-601 Rzeszów, Poland; pczarnota1@gmail.com

4  Scientific Laboratory of the Gorce National Park, Poręba Wielka 590, 34-735 Niedźwiedź, Poland

*  Correspondence: agata.kaczmarek@upwr.edu.pl; Tel.: +48-713201711

**Abstract:** The aim of the research was to check whether the healing of bark-stripping wounds of the silver fir tree trunks reduces the share of wood-decomposing fungi, which may be the result of inter-species interactions. The study carried out in Gorce National Park in Polish Western Carpathians analyzed drill holes of sapwood from three types of wounds (fresh, healed and old) on fir trunks with a diameter at breast height (DBH) of 4.0–16.9 cm as a result of bark-stripping by red deer (*Cervus elaphus*). In the wood of fresh wounds *Alternaria alternata* (Fr.) Keissl. and *Arthrinium arundinis* (Corda) Dyko & B. Sutton had the largest share in mycobiota. *Phompsis* spp. and the species *Sydowia polyspora* (Bref. & Tavel) E. Müll. and *Epicoccum nigrum* Link were also isolated. The dominants in old wounds were *Eutypa* spp., *Phomopsis* spp. and *Cylindrobasidium evolvens* (Fr.) Jülich. Healed wounds were dominated by *Trichoderma atroviride* P. Karst, a fungus antagonistic to many fungal pathogens. Such properties are shared by *A. arundinis*, especially common in fresh wound wood. It seems that these fungi support the process of wounded tree regeneration (healing of wounds) and limit the activity of wood-decaying fungi in old age, which makes fir survival very high. Thus, even a strong red deer pressure cannot be considered the basic factor determining the dynamics of fir in this part of the Carpathians.

**Keywords:** forest ecology; fungal ecology; mountain fir-spruce forest; wood decay fungi; bark stripping; *Abies alba*; ITS rDNA; barcoding of fungi; ungulates; red deer; Gorce National Park; Carpathians

## 1. Introduction

The bark stripping by ungulates is the result of their need for food and specific nutrients contained in the bark, including different fiber fractions [1] in the hard-to-survive winter period [2], but also in summer [3]. This method of a diet supplementing is typical for forest representatives of Cervidae including genera *Alces*, *Capreolus*, *Cervus*, *Dama*, *Odocoileus*, *Rangifer* [1] and, unless it causes direct, serious economic damage to forests, is generally acceptable and these animals are treated as key factors of natural ecological processes shaping inter-species relationships and the diversity and structure of forest plant communities [4,5]. Open wounds, however, allow the infection of wood-inhabiting fungi, the presence of which is already most often perceived negatively in commercial forests.

The problem of young tree bark-stripping and then wound colonization, which in turn may lead to infection of tree tissues by pathogenic fungi, has long been noticed by

researchers. The first Polish reports about the significant impact of bark-stripping by red deer (*Cervus elaphus*) on the infection of trees by fungi come from the Karkonosze Mountains in the 1960s. Domański [6] found that open wounds on spruce trunks facilitate infection by saprophytes and pathogens. Among the potential perpetrators of wood decay, he mentions fungi of the genus *Stereum*, but also species currently included in the so-called *Deuteromycota* fungi. His results were later confirmed by many researchers, who isolated many dangerous pathogens from bark-stripped spruce trunks, such as *Heterobasidion* spp. [7]. The study also confirmed the presence of the species of fungi that were then classified as *Deuteromycota*, and which are responsible for the so-called gray or mildew rot of wood [8]; their role may be much more important in the process of wood decomposition than previously thought [7,8].

Arhipova et al. [9] studied, among others, the species composition of fungi occurring in *Pinus contorta* Douglas ex Loudon stem wounds, resulting from elk and deer bark-stripping. Researchers isolated a total of 28 taxa of fungi found in more than half of the wood samples tested. *Ascomycota* anamorphs was the dominating species. However, studies by many authors reveal different sensitivity of individual tree species to bark-stripping and, consequently, to fungal infections and the appearance of wood rot. In the case of fir, the species composition of fungi inhabiting wounds has not been studied so far [10].

Silver fir (*Abies alba Mill.*) is the sixth most productive forest-forming species in Europe [11]. Its native distribution ranges from the Pyrenees in the west to the Carpathians in the east and from the Polish lowlands in the north to southern Italy in the south [12]. In the Carpathians, it is one of the three main forest-forming species, next to beech and spruce. It is most numerous in the *Abieti-Piceetum montanum* fir-spruce forests and in the fir variant of the Carpathian beech forest *Dentario glandulosae-Fagetum abietetosum* [13]. The situation is similar in Gorce, in the Polish Western Carpathians [14,15], where Gorce National Park was established in 1981 on an area of just over 7000 ha. Its strategic goal is to preserve or restore natural features to forests through strict protection of spontaneous natural processes in the ever-larger territory of the park, currently implemented on an area of 4009 ha. One such process is red deer pressure on forests, which results in damaging the young generation of silver fir by browsing and bark-stripping [16,17]. This phenomenon is generally perceived negatively by many people, especially by foresters, hunters and locals sharing space with wild herbivorous [18,19]. The pressure of these animals on forests is also controversial for nature protection services themselves, which often identify their mission with the need to protect the forest using management methods and to treat fir as an endangered species that requires special care [20]. This view is largely the result of the phenomenon of mass extinction of fir in Europe in the second half of the last century [21,22], and the unique food preference of this forest-forming species by deer [20,23,24]. It is also supported by the usually locally poor regeneration capacity of fir stands, which, due to climatic, geomorphological and lighting conditions, can be highly variable and irregular in time and space [25–27], depending on the occurrence of the 'rare windows of opportunity' for fir [23].

The conviction about the poor condition of the fir, despite the noble cause of wildlife preservation, is the reason for the so-called active protection measures in Polish Carpathian national parks, consisting in securing treetops with plastic spirals, and trunks of young firs with repellents [20]. Even if data from various regions of the Carpathians indicate that fir is currently in expansion [25,28]; that the condition of fir trees which underwent bark-stripping, expressed in terms of growth rate and tree vitality, does not significantly differ from the "normal" ones [10]; and that the survival rate of the young generation of this species, despite intensive gnawing, is very high [20,29], there are still worries about its health. The perspective of future forests, including natural ones, in which bark-stripped fir trees will not grow old due to the destructive fungi developing in the wood, raises concerns. These fears, as well as the desire to learn about the diversity of fungi inhabiting the wood of fallen firs in different thickness classes, their phytopathological significance

and, finally, possible inter-species interactions in the communities of these fungi, became the reason for conducting this study.

The working hypotheses that we wanted to verify are: (1) the type of bark stripping wound and DBH of trunk determine wood inhabiting fungal diversity and the share of wood-decomposing fungi, and (2) the natural reduction in the share of wood-decomposing fungi in healing silver fir trunks is due to fungal species interactions inhabiting wounds, which results in high regenerative capacities of this forest-forming species. The last hypothesis was based on the results of the study by Pach [10], who found that with age, there is a significant decrease in the rate of decay and spread of wood discoloration of bark-stripped fir trees.

The obtained results may provide the answer to the following pragmatic question: whether in forest habitats dominated by fir (or in the reconstruction phase), particularly in national parks, it is necessary to apply cost-intensive, mechanical (plastic spirals for apical shoots) and chemical (repellents) protection of this forest-forming species against red deer. It is all the more important as, in the time of climate change and the progressive decay of mountain spruce stands, fir seems to be a species of the future [13,25].

## 2. Materials and Methods

### 2.1. Fieldwork

The research was carried out in the Gorce National Park (Western Carpathians, Southern Poland) within the active protection zone covered with young (early optimal growth stage) fir-spruce forest of the lower belt *Abieti-Piceetum montanum* (Figure 1) extending on acid brown soil. Wood samples were taken with a Pressler drill from a sapwood part of silver fir trunks, referring to three categories of wounds: fresh wounds marked with the symbol "F" (Figure 2A), healed wounds marked with the symbol "H" (Figure 2B) and old, and open wounds marked with the symbol "O" (Figure 2C). The drill was decontaminated with Aerodesin 2000 before each drilling.

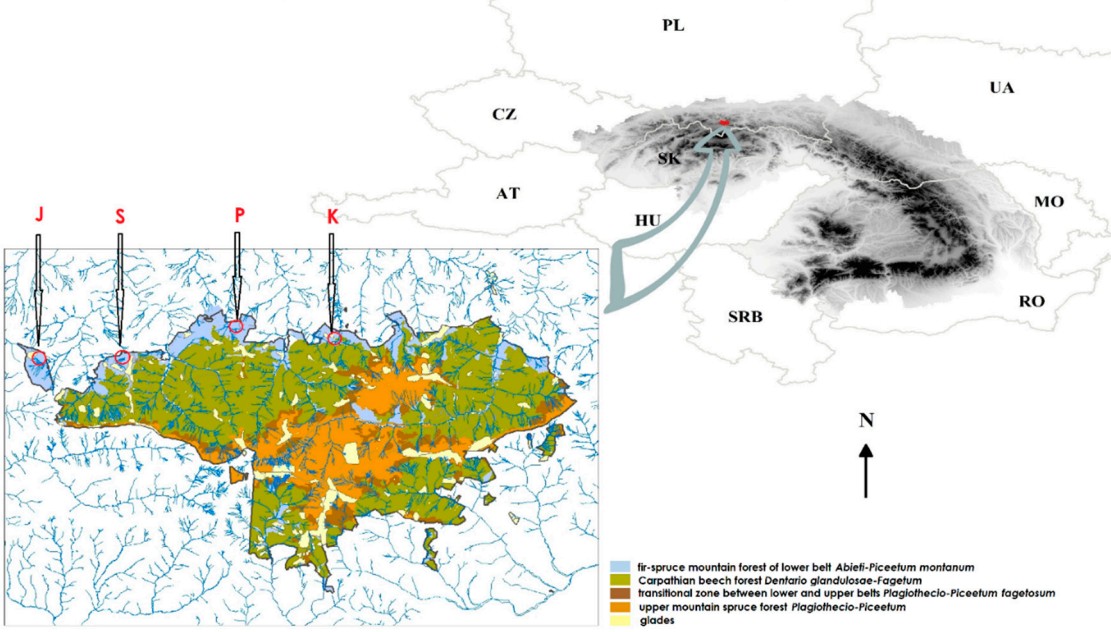

**Figure 1.** Localities of sampled forest area within fir-spruce mountain forest *Abieti-Piceetum montanum* in the Gorce National Park (W Carpathians, S Poland). Abbreviations: J—Suhora forest district, Jasionów forest area, K—Kudłoń forest district, Za Palacem forest area, P—Kudłoń forest district, Pasieka forest area, S—Suhora forest district, Czarny Groń forest area.

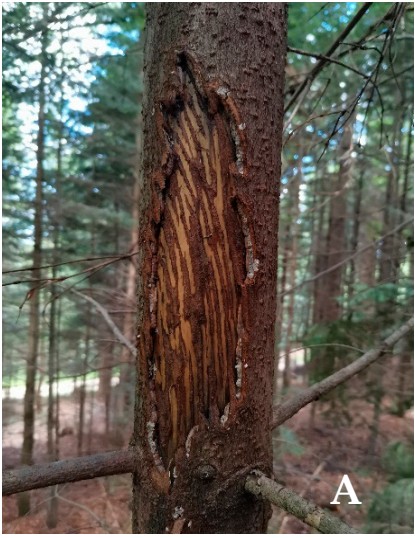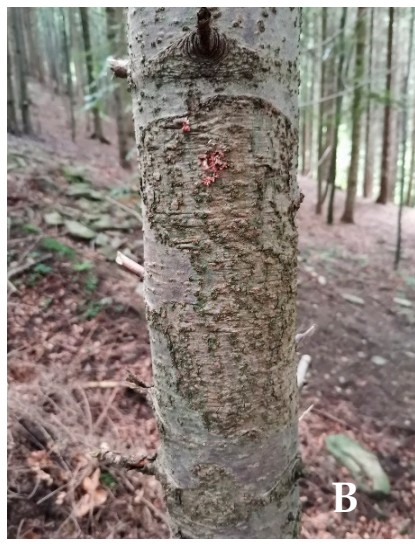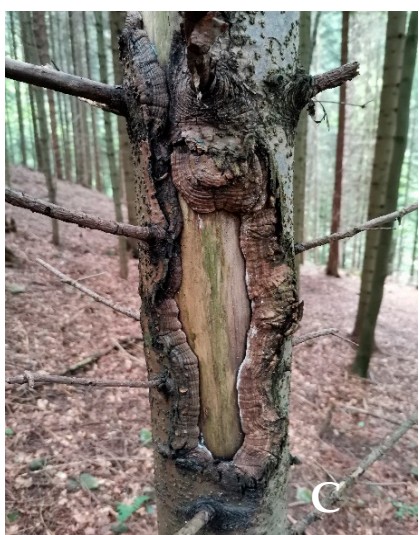

**Figure 2.** (**A**–**C**) Types of studied fir wounds caused by red deer (*Cervus elaphus*): (**A**)—fresh wound "F", (**B**)—healed wound "H", (**C**)—old wound "O".

The study covered trees of various age classes, corresponding to the following values of a diameter at breast height (DBH): 4.0–6.9 cm, 7.0–9.9 cm, 10.0–12.9 cm, 13.0–16.9 cm. The sole criterion for selecting the sample trees was that they had achieved the established thickness class DBH and wound type, so that each wound type in each thickness class was represented by ± 10–12 trees. This choice was firstly determined by the shortest distance between the fir tree and the linear temporal transect in each of the four heavily bark stripped regions, and in the absence of trees meeting the criteria, they were selected randomly from the immediate vicinity. A total of 141 wood samples from bark-stripping wounds were collected (Table 1). Additionally, wood samples were collected in the same way from 11 firs with DBH between 30.0–40.0 cm with no visual signs of bark-stripping, growing in the vicinity of damaged trees. These trees were taken as a reference sample, assuming that in the past they were also subjected to similar pressure of red deer. It was assumed that the healing of wounds and further growth of these trees is the result of high regenerative abilities of this tree species, resulting from the natural mechanism allowing them to neutralize pathogenic fungi inhabiting wounds sustained at younger age.

**Table 1.** The number of collected drill samples of sapwood in reference to the category of bark-stripping wounds of *Abies alba* stems.

| DBH [cm] | Fresh Wounds | Healed Wounds | Old Wounds |
|---|---|---|---|
| 4.0–6.9 | 13 | 13 | 12 |
| 7.0–9.9 | 11 | 12 | 12 |
| 10.0–12.9 | 12 | 15 | 12 |
| 13.0–16.9 | 5 | 10 | 14 |
| Sum | 41 | 50 | 50 |
| Control > 30.0 | | 11 | |
| In total | | 152 | |

### 2.2. Laboratory Analysis

The pieces of wood were placed on MEA medium (Malt Extract Agar, Difco, La Vista, NE, USA) in accordance with the methodology proposed by Arhipova et al. [9] and PDA medium (Potato Dextrose Agar, Difco, Lawrence, KS, USA). Growing colonies of fungi were passaged and then clean mycelium was obtained by the method of monospore cultures, meeting the requirements of the molecular method of determining species affiliation.

Isolates grown in Petri dishes with PDA, showing morphological diversity, and representing various taxonomic units with high probability, were used for molecular analyses based on the sequence of ITS rDNA regions. The subject of molecular identification were representatives separated on the basis of microscopic morphological features suggesting their belonging to different species/genera. The aim of the analysis of ITS regions, in addition to the identification of the analyzed isolates, was to graphically depict the genetic connections between isolates and enrich the NCBI international database with their sequences. From pure cultures of individual isolates grown on PDA, three mycelial discs 5 mm in diameter were cut and transferred to Petri dishes with PDB glucose-potato liquid medium (Potato Dextrose Broth, A&A Biotechnology, Gdańsk, Poland), and protected with parafilm. The mycelium obtained after 5–7, and in the case of the slowest growing specimens—after 10 days, was rinsed with sterile water, filtered using a filtering set consisting of filter paper, a Büchner funnel and a vacuum pump, and then lyophilized for 2 days in a Cool Safe freeze dryer (Scanvac, Lynge, Denmark). DNA isolation was performed according to the modified method of Doyle and Doyle [30]. Thirty milligrams of mycelium of each isolate homogenized in a Magna Lyser homogenizer (Roche, Basel, Switzerland) using quartz beads and quartz sand was placed in a 2 mL tube and covered with 900 μL of extraction buffer containing CTAB 5.0%, EDTA 0.5 M, NaCl 5.0 M, Tris -HCl (pH 8.0) 1.0 M, 2-mercaptoethanol and PVP 2.0%. After incubation at 65 °C, phenol, chloroform and isoamyl alcohol were used to remove proteins and carbohydrates. In the following stages, 95% and 70% ethyl alcohol were used. The obtained DNA was suspended in 200 μL of ddH2O and cleaned with an anti-inhibitor kit (A&A Biotechnology, Gdańsk, Poland).

DNA was measured fluorometrically on a Quantus device (Promega, WI, USA) and diluted in $ddH_2O$ to a concentration of 10 ng·μL$^{-1}$ for further analysis. The PCR reaction aimed to amplify the ITS regions was performed in a volume of 37.5 μL containing the PCR Core Kit reagents (QIAGEN, Germantown, MD, USA): 1x buffer, 1x Q solution, 1 mM $MgCl_2$, 0.2 mMdNTP, 0.6 pM of each of the two primers (ITS1: 5′-TCCGTAGGTGAACCTGCGG-3′ and ITS4: 5′-TCCTCCGCTTATTGATATGC-3′, White et al. [31] and DNA with a concentration of 10 10 ng·μL$^{-1}$. DNA was amplified in an Eppendorf EP Mastercycler according to the reaction protocol: pre-denaturation 3 min at 95 °C—3 min, 35 cycles (95 °C—1 min, 55 °C—45 s, 72 °C—1 min) and final elongation at 72 °C—10 min. The presence of reaction products was verified after electrophoretic separation in TBE buffer, carried out on a 1.2% agarose gel (Pronadisa, Spain) with the addition of Simply Safe (EURx, Poland) dye, applying 2 μL of the post-reaction mixture.

The amplification products were purified and sequenced by Genomed (Poland). FinchTV 1.4 (Geospiza Inc., WA, USA) was used to analyze the sequences obtained. ClustalW analysis was performed on Mega7 Toolbar [32] (accessed on 10 October 2020). The Basic Local Alignment Search Tool (BLAST) in the NCBI database (The National Center for Biotechnology Information) [33] (accessed on 10 October 2020) was used to identify species based on the ITS sequence. The Mega7 Toolbar tools [32] were used to determine the sequence differentiation (overall mean distance) and dendrogram construction based on the ITS rDNA sequence using the maximum likelihood algorithm (MLE) based on the Kimura 2-parameter model [34,35].

The nomenclature of the identified taxa is in accordance with the Catalogue of Life [36] (accessed on 10 October 2020) and the Index Fungorum [37] (accessed on 10 October 2020).

### 2.3. Statistical Analysis

Based on the achieved results the differences in the abundance of wood decaying fungi inhabiting three types of bark-stripping wounds (plus control) have been shown on a box-plot graph. The same type of graph was used to show differences in the fungal diversity of these wounds. For this purpose, the three α-diversity indexes were calculated: Shannon-Wiener (H), Pielou (J) and Simpson (D), based on mathematical equation presented in Patejuk and Pusz [38]. β-biodiversity Sørensen index was calculated separately. Moreover, the linear regression between fungal colony abundance on types of bark wounds and the

number of annual rings and DBH was calculated. Graphical representations and basic statistics were made in Tableau (2020.2.4 Professional Edition).

## 3. Results

### 3.1. Taxonomy and Frequency of Fungal Colonies

A total of 41 taxa of fungi were isolated and assigned to a species or higher systematic rank (Table 2, Figure 3). *Trichoderma atroviride* P. Karst and the fungi of the genus *Eutypa* were dominant among all the isolated fungal colonies, reaching the frequency of nearly 24% and 13%, respectively. There were also relatively many colonies of such taxa as: *Phomopsis* spp. (10.19%), *Cylindrobasidium evolvens* (Fr.) Jülich (8.57%), *Paraphaeosphaeria neglecta* Verkley, Riccioni & Stielow (8.29%), *Arthrinium arundinis* (Corda) Dyko & B. Sutton (7.71%), and *Alternaria alternata* (Fr.) Keissl. (6.67%). The level of remaining taxa colonizing bark-stripping wounds ranged from 3.4 to 0.1% (Figure 4).

**Table 2.** Fungi isolated from sapwood of bark-stripped *Abies alba*. Molecular identification based on ITS rDNA region sequences using BLAST method and maximum likelihood ITS phylogeny.

| Fungal Taxa | Number | Frequency [%] | Accession Number in NCBI | Identity with NCBI Isolates (Accession Number) |
|---|---|---|---|---|
| *Alternaria alternata* (Fr.) Keissl. | 70 | 6.67 | MW090865 | MT644140, MT487778 |
| *Arthrinium arundinis* (Corda) Dyko & B. Sutton | 81 | 7.71 | MW090861 MW090870 MW113167 MW090903 MW113168 MW113226 MW113227 MW113228 | MT582801, MT446201 |
| *Aureobasidium pullulans* (de Bary & Löwenthal) G. Arnaud | 4 | 0.38 | MW090923 | MT363099, MN922125 |
| *Botrytis cinerea* Pers. | 8 | 0.76 | MW090881 | MT573470, MN448502 |
| *Cadophora* sp. | 23 | 2.19 | MW090880 MW090922 | MF782737, MF188972 |
| *Chaetomium* sp. | 2 | 0.19 | MW090810 | MH171491, KC963908 |
| *Coniochaeta* sp. | 1 | 0.10 | MW090815 | MH859071, MG905629 |
| *Coprinellus micaceus* (Bull.) Vilgalys, Hopple & Jacq. Johnson | 3 | 0.29 | MW090910 | MH179313, GU227721 |
| *Cucurbitariaceae* sp. | 1 | 0.10 | MW090913 | KC963916, MK460387 |
| *Cylindrobasidium evolvens* (Fr.) Jülich | 90 | 8.57 | MW113169 | MN947592, MH854673 |
| *Cystobasidium larynges* (Reiersöl) Yurkov, Kachalkin, H.M. Daniel, M. Groenew., Libkind, V. de García, Zalar, Gouliam., Boekhout & Begerow | 1 | 0.10 | MW090823 | MH047192, KY103134 |
| *Cytospora* sp. | 19 | 1.81 | MW090820 MW090909 | KY051899, KU516449 |
| *Epicoccum nigrum* Link | 34 | 3.24 | MW090813 | MT548679, LC543647 |
| *Eutypa* sp. | 129 | 12.29 | MW090805 MW090860 MW090912 MW090911 | AY620998, KF453561 |
| *Fusarium acuminatum* Ellis & Everh. | 6 | 0.57 | MW090924 | MT649858, MT635295 |
| *Fusarium avenaceum* (Fr.) Sacc. | 1 | 0.10 | MW090925 | MT446118, MT276139 |
| *Fusarium tricinctum* (Corda) Sacc. | 15 | 1.43 | MW090875 MW090907 MW090918 | MK934343, KC311496 |
| *Helotiales* sp. | 1 | 0.10 | MW090817 | DQ317330, MF494618 |
| *Heterobasidion annosum* (Fr.) Bref. | 1 | 0.10 | MW090819 | MK395162, KU727784 |
| *Neonectria neomacrospora* (C. Booth & Samuels) Mantiri & Samuels | 5 | 0.48 | MW090919 | MH580206, MG049669 |
| *Nigrospora oryzae* (Berk. & Broome) Petch | 9 | 0.86 | MW090868 | MT556421, MG661721 |
| *Paraphaeosphaeria neglecta* Verkley, Riccioni & Stielow | 87 | 8.29 | MW090864 MW090874 MW090905 MW090906 | MK646057, MN244542 |
| *Penicillium chrysogenum* Thom | 17 | 1.62 | MW090830 | MT524448, MK762610 |

**Table 2.** *Cont.*

| Fungal Taxa | Number | Frequency [%] | Accession Number in NCBI | Identity with NCBI Isolates (Accession Number) |
|---|---|---|---|---|
| *Peniophoraceae* sp. | 6 | 0.57 | MW090901 | MH010048, MH857634 |
| *Phomopsis* sp. | 107 | 10.19 | MW090829 MW090876 | MN538335, MT877030 |
| *Preussia minima* (Auersw.) Arx | 3 | 0.29 | MW090811 | MG457827, MT645911 |
| *Rhizosphaera macrospora* Gourb. & M. Morelet | 4 | 0.38 | MW090806 | AM884745, MN538337 |
| *Rhizosphaera oudemansii* Maubl. | 1 | 0.10 | MW090827 | KU516578, EU700366 |
| *Sarea difformis* (Fr.) Fr. | 5 | 0.48 | MW090809 | MN699648, FR837921 |
| *Schizophyllum commune* Fr. | 9 | 0.86 | MW113225 | MT647523, MT601951 |
| *Sordaria fimicola* (Roberge ex Desm.) Ces. & De Not. | 1 | 0.10 | MW090812 | MK965099, KX986578 |
| *Stereum sanguinolentum* (Alb. & Schwein.) Fr. | 6 | 0.57 | MW090831 | AY618670, AF533962 |
| *Sydowia polyspora* (Bref. & Tavel) E. Müll. | 28 | 2.67 | MW090808 MW090828 MW090873 | KY659505, KU837235 |
| *Tolypocladium* sp. | 2 | 0.19 | MW090807 | MN096582, MH730171 |
| *Trametes versicolor* (L.) Lloyd | 2 | 0.19 | MW090818 MW090825 | KJ995921, EU661891 |
| *Trametes* sp. | 3 | 0.29 | MW090814 | MK343672, MK269115 |
| *Trichoderma atroviride* P. Karst | 248 | 23.62 | MW090869 | MT341775, MT023026 |
| *Truncatella angusta* (Pers.) S. Hughes | 5 | 0.48 | MW090908 | MT514378, MK647988 |
| *Valsa* sp. | 1 | 0.10 | MW090826 | HQ654894, KY051908 |
| *Xylariales* sp. | 10 | 0.95 | MW090804 MW090816 MW090821 MW090822 MW090878 | KC774617, KF415082 |
| *Zalerion* sp. | 1 | 0.10 | MW090824 | AY465470, AF169308 |

### 3.2. Fungi-Wounds Relationship

When analyzing the obtained results, it can be concluded that the most fungal colonies were isolated from old wounds (434 colonies and 17 taxa). For fresh and scarred wounds, this number was 249 (18 taxa) and 212 (24 taxa), respectively, compared to the total number of 154 (nine taxa) colonies obtained from the control drills (Table 3). In the case of fresh wounds, species such as *A. alternata* and *A. arundinis* had the highest share. Fungi belonging to the genus *Phomopsis* were also isolated, as well as the species *Sydowia polyspora* (Bref. & Tavel) E. Müll. and *Epicoccum nigrum* Link. In the case of fresh wounds, *T. atroviride* was a distinct dominant, which also dominated in the mycobiota of wood collected from control trees. On the other hand, in the case of old wounds, the fungi of the genera *Eutypa* and *Phomopsis*, as well as the fungus *C. evolvens*, were dominant.

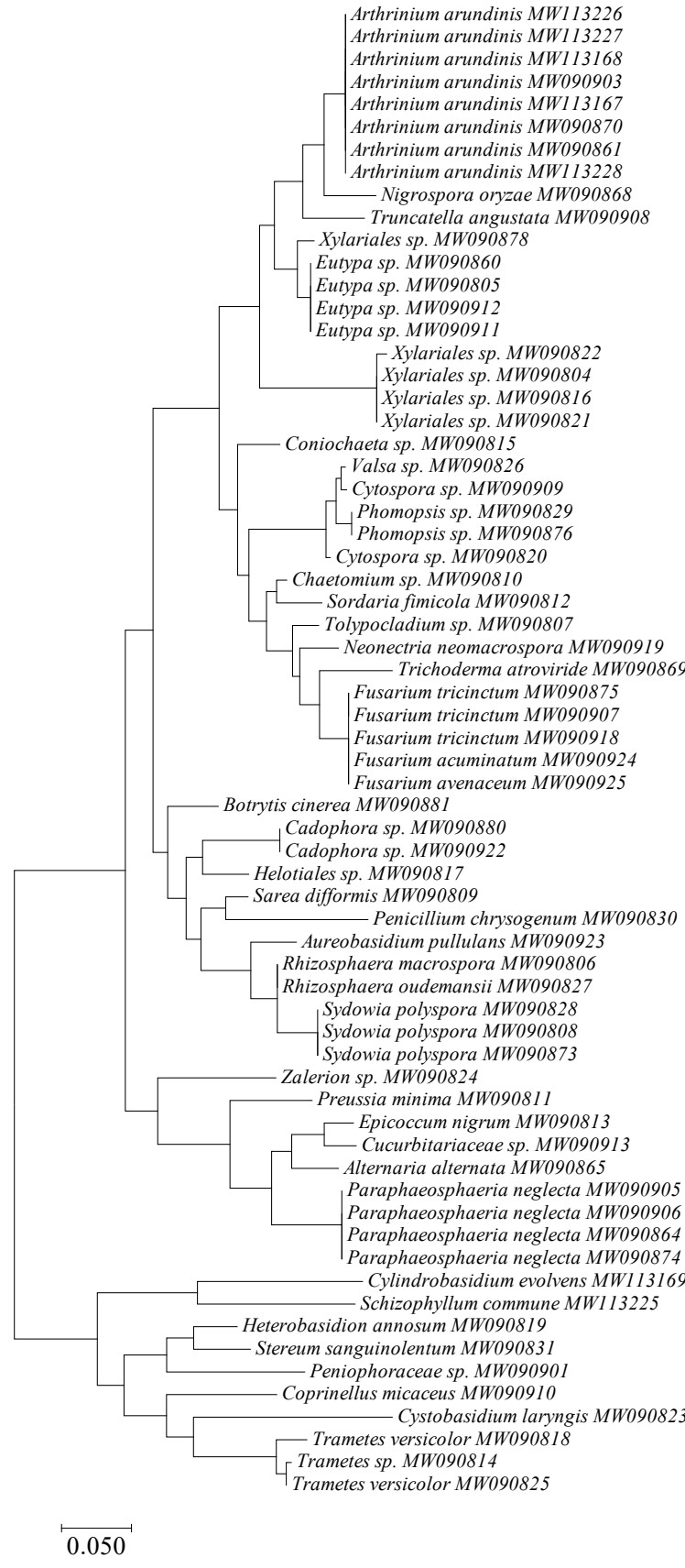

**Figure 3.** A dendrogram based on the sequencing of ITS regions in MEGA7 software, depicting the level of genetic similarity between 66 isolates obtained from silver fir wood in the Gorce National Park in 2020.

**Table 3.** Fungi colonies isolated from sapwood of bark-stripped *Abies alba* firs depending on wound type and habitation depth (O—outside part of wood; M—middle part of wood; H—the part near heartwood).

| Fungal Taxa | Wound | | | | | | | | | Control | | |
|---|---|---|---|---|---|---|---|---|---|---|---|---|
| | Fresh | | | Healed | | | Old | | | | | |
| | **O** | **M** | **H** | **O** | **M** | **H** | **O** | **M** | **H** | **O** | **M** | **H** |
| *Alternaria alternata* | 14 | 6 | 9 | 6 | 6 | 1 | 11 | 12 | | | 5 | |
| *Arthrinium arundinis* | 19 | 12 | 9 | 4 | 3 | 1 | 3 | 14 | 7 | 6 | | 3 |
| *Aureobasidium pullulans* | | | 1 | | | 3 | | | | | | |
| *Botrytis cinerea* | 3 | 1 | | | | | 3 | 1 | | | | |
| *Cadophora* sp. | | | | | | | 6 | 11 | 6 | | | |
| *Chaetomium* sp. | | | | | | 2 | | | | | | |
| *Coniochaeta* sp. | | | | 1 | | | | | | | | |
| *Coprinellus micaceus* | | | 3 | | | | | | | | | |
| *Cucurbitariaceae* sp. | | | | | | | | | 1 | | | |
| *Cylindrobasidium evolvens* | 7 | 6 | 6 | | | | 23 | 24 | 24 | | | |
| *Cystobasidium laryngis* | | | | | | 1 | | | | | | |
| *Cytospora* sp. | 6 | 6 | 6 | 1 | | | | | | | | |
| *Epicoccum nigrum* | 9 | 6 | 11 | | | | 5 | | | | | 3 |
| *Eutypa* sp. | | | | | | | 51 | 42 | 36 | | | |
| *Fusarium acuminatum* | | | | | | | | 6 | | | | |
| *Fusarium avenaceum* | | 1 | | | | | | | | | | |
| *Fusarium tricinctum* | | | | 3 | 6 | | 1 | 2 | | | 3 | |
| *Helotiales* sp. | | | | | | 1 | | | | | | |
| *Heterobasidion annosum* | | | | | | 1 | | | | | | |
| *Neonectria neomacrospora* | | | | | | 5 | | | | | | |
| *Nigrospora oryzae* | | 6 | 3 | | | | | | | | | |
| *Paraphaeosphaeria neglecta* | 2 | 6 | 6 | | | 6 | 1 | 12 | 18 | 12 | 12 | 12 |
| *Penicillium chrysogenum* | | | | | | | 1 | 4 | 6 | | 6 | |
| *Peniophoraceae* sp. | | | | | | | | | | | | |
| *Phomopsis* sp. | 7 | 10 | 15 | | 3 | 1 | 37 | 20 | 12 | | | 2 |
| *Preussia minima* | 2 | | | | | | 6 | | | | | |
| *Rhizosphaera macrospora* | | | | 4 | | | | | | | | |
| *Rhizosphaera oudemansii* | | | | 1 | | | | | | | | |
| *Sarea difformis* | | | | | | 5 | | | | | | |
| *Schizophyllum commune* | | | | | | | | | | 1 | 2 | 6 |
| *Sordaria fimicola* | | | | | | 1 | | | | | | |
| *Stereum sanguinolentum* | | | 6 | | | | | | | | | |
| *Sydowia polyspora* | 8 | 4 | 8 | 5 | 1 | 1 | 1 | | | | | |
| *Tolypocladium* sp. | | | | | 1 | 1 | | | | | | |
| *Trametes versicolor* | | | | 1 | | 1 | | | | | | |
| *Trametes* sp. | | | | | | 3 | | | | | | |
| *Trichoderma atroviride* | 6 | 6 | 7 | 42 | 42 | 43 | 6 | 8 | 7 | 24 | 27 | 30 |
| *Truncatella angustata* | 2 | | 3 | | | | | | | | | |
| *Valsa* sp. | | | | 1 | | | | | | | | |
| *Xylariales* sp. | 1 | | | 1 | 1 | 1 | 4 | 1 | 1 | | | |
| *Zalerion* sp. | | | | | | 1 | | | | | | |
| In total: | 86 | 70 | 93 | 70 | 63 | 79 | 159 | 157 | 118 | 43 | 55 | 56 |

The number of colonies isolated from the samples was highly variable depending on the sample, but the median for the examined wounds was 0 for fresh wounds, 1 for healed wounds and 7 for old wounds (Figure 5). Control samples had the highest number of colonies, and in this case the distribution of the number of colonies isolated from individual samples had the features of normal distribution.

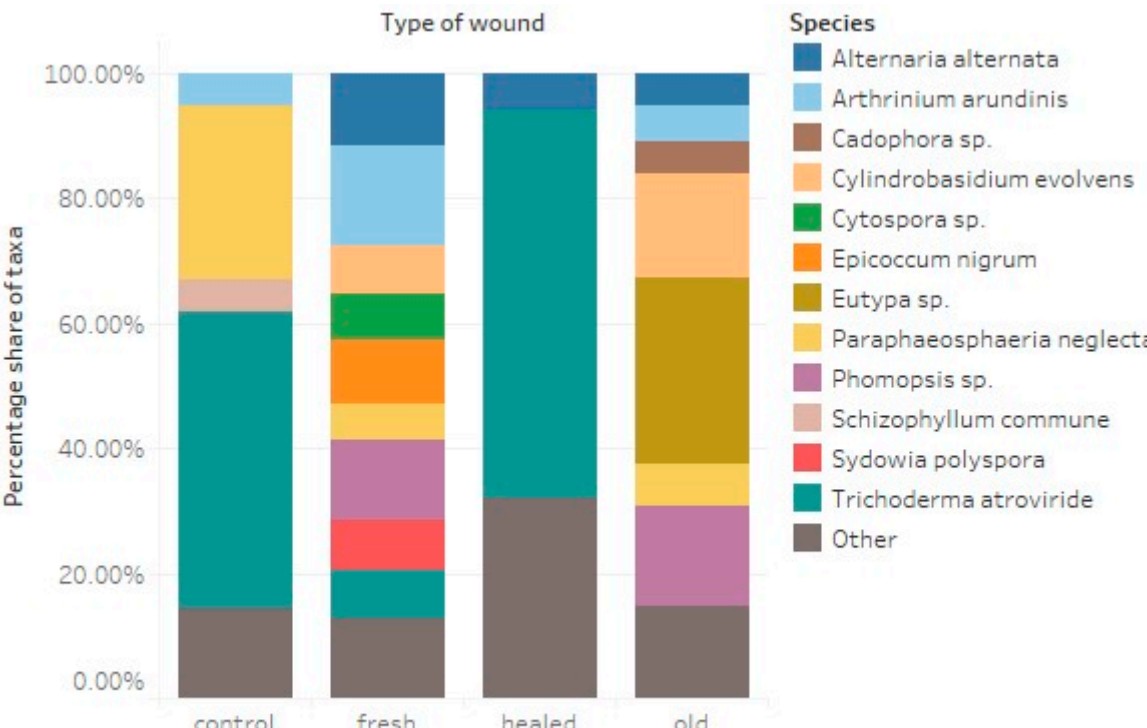

**Figure 4.** Percentage share of fungal taxa inhabiting wood of three types of wounds caused by red dear on silver fir trunks and wood of control fir trees (species constituting less than 5% in total of fungal colonies were grouped to "Other").

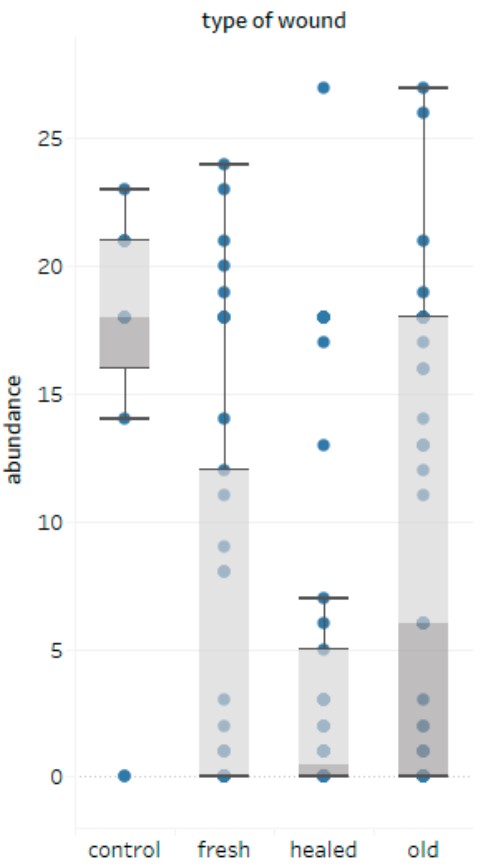

**Figure 5.** Box-plot representation of the abundance of fungal colonies in the wood sample, according to the type of wound on silver fir trunk.

When analyzing wounds on fir trunks in terms of their DBH, it was found that the largest number of colonies occurs in thinner trees (DBH 4.0–6.9 cm). In their case, 306 colonies belonging to 25 taxa were obtained. In trees thicker than 7.0 cm, the number of colonies ranged from 113 to 271, with 32 taxa. The fungi of the genus *Eutypa* and *Phomopsis* had the highest share (Table 4).

**Table 4.** The number of isolated fungi colonies inhabiting wounds depending on the DBH of bark-stripped firs.

| Fungal Taxa | DBH [cm] | | | | |
|---|---|---|---|---|---|
| | 4.0–6.9 | 7.0–9.9 | 10.0–12.9 | 13.0–16.9 | Control |
| *Alternaria alternata* | 43 | 6 | 4 | 12 | 5 |
| *Arthrinium arundinis* | 23 | | 19 | 30 | 9 |
| *Aureobasidium pullulans* | | 3 | | 1 | |
| *Botrytis cinerea* | 4 | 2 | 1 | 1 | |
| *Cadophora* sp. | | | 12 | 11 | |
| *Chaetomium* sp. | | 2 | | | |
| *Coniochaeta* sp. | 1 | | | | |
| *Coprinellus micaceus* | 3 | | | | |
| *Cucurbitariaceae* sp. | | 1 | | | |
| *Cylindrobasidium evolvens* | 33 | 33 | 18 | 6 | |
| *Cystobasidium laryngis* | | 1 | | | |
| *Cytospora* sp. | 18 | | 1 | | |
| *Epicoccum nigrum* | 7 | | 6 | 18 | 3 |
| *Eutypa* sp. | 42 | 24 | 33 | 30 | |
| *Fusarium acuminatum* | | | 6 | | |
| *Fusarium avenaceum* | | | | 1 | |
| *Fusarium tricinctum* | 1 | | 2 | 9 | 3 |
| *Helotiales* sp. | 1 | | | | |
| *Heterobasidion annosum* | | 1 | | | |
| *Neonectria neomacrospora* | | | | 5 | |
| *Nigrospora oryzae* | 9 | | | | |
| *Paraphaeosphaeria neglecta* | 20 | | 23 | 6 | 48 |
| *Penicillium chrysogenum* | 1 | | 10 | | 6 |
| *Peniophoraceae* sp. | | | | | 6 |
| *Phomopsis* sp. | 17 | 19 | 41 | 28 | 2 |
| *Preussia minima* | | | 2 | 6 | |
| *Rhizosphaera macrospora* | 2 | | | 2 | |
| *Rhizosphaera oudemansii* | 1 | | | | |
| *Sarea difformis* | 5 | | | | |
| *Schizophyllum commune* | | | | | 9 |
| *Sordaria fimicola* | 1 | | | | |
| *Stereum sanguinolentum* | | 6 | | | |
| *Sydowia polyspora* | 12 | 10 | 1 | 5 | |
| *Tolypocladium* sp. | | | | 2 | |
| *Trametes versicolor* | 1 | 1 | | | |
| *Trametes* sp. | | | 3 | | |
| *Trichoderma atroviride* | 54 | 1 | 33 | 91 | 81 |
| *Truncatella angustata* | 5 | | | | |
| *Valsa* sp. | 1 | | | | |
| *Xylariales* sp. | 1 | 2 | | 7 | |
| *Zalerion* sp. | | 1 | | | |
| In total: | 306 | 113 | 215 | 271 | 172 |

The impact of the measured parameters of trees, i.e., the number of annual rings and DBH, on the number of fungal colonies isolated from sapwood drillholes from individual types of wounds is interesting (Figure 6). The number of isolates from old wounds was not significantly influenced by either the number of annual rings (age of the trees) or the DBH of firs. A very weak positive correlation at the level of $R^2 = 0.01229$ was found between the number of colonies inhabiting scarred wounds and the age of the trees, while the DBH of

the sampled firs was not significant. Significant differences were found in the case of fresh wounds, where the number of isolated colonies decreased with the increasing number of annual rings and DBH. The detected correlation was average and depending on the parameter was $R^2 = 0.2015$ for annual rings and $R^2 = 0.1205$ for DBH.

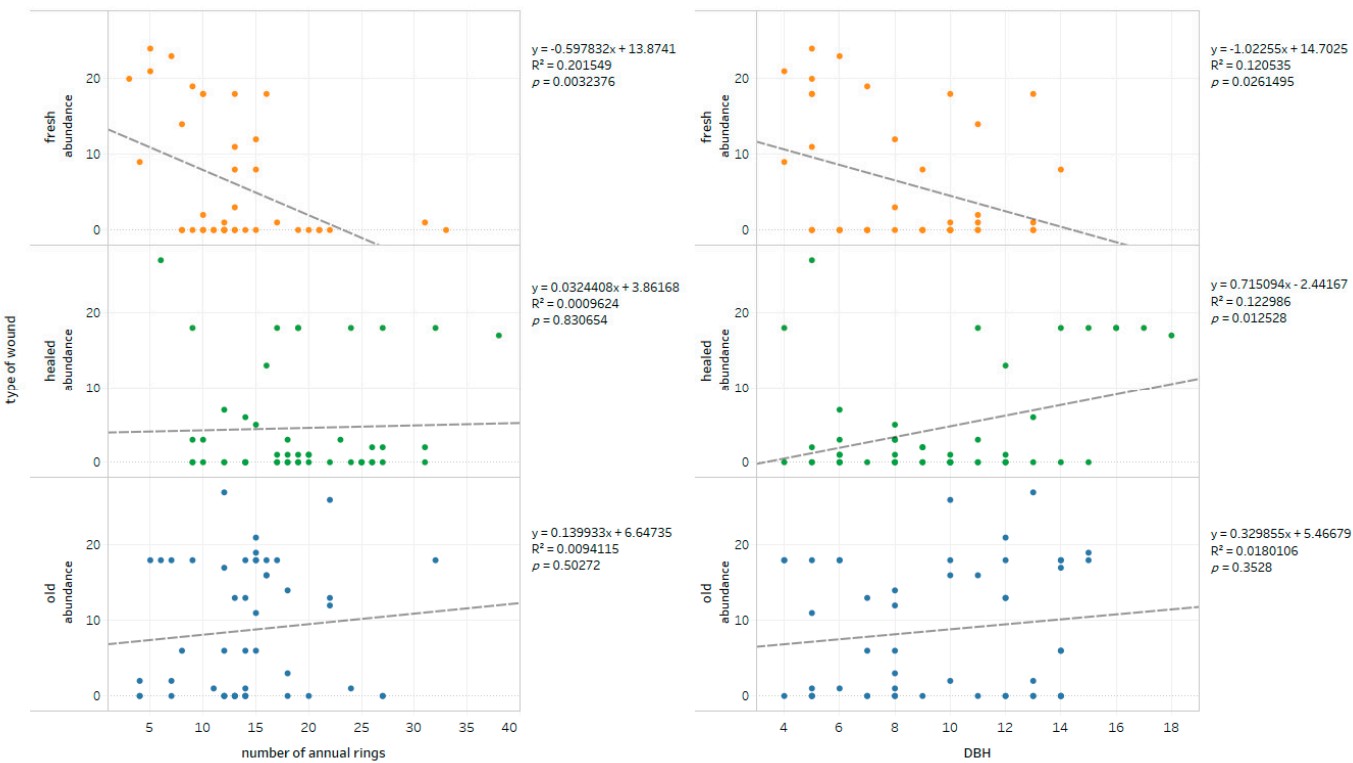

**Figure 6.** Linear regression of fungal colony abundance on three types of bark wounds against the number of annual rings and DBH of investigated silver fir trunks.

### 3.3. Wounds-Fungal Diversity Relationship

On the basis of the obtained results for samples with all variables taken into account (location, type of wound and part of wood), α- and β-biodiversity indexes were calculated, describing the differences between the studied populations. Detailed results are presented in Tables 5 and 6. If no isolates were identified from a given sample, they were not included in the biodiversity analysis (Table 5). Samples of very poor biodiversity (1–2 taxa in the sample), which do not meet the requirements for calculations, are marked with "0" values (Table 5).

The most common values of the Shannon-Wiener and Simpson diversity indices for all trials with healed wounds were lower than the values of these indices for fresh and old wounds (Figure 7). The range of the Pielou evenness index values was much wider for the mycobiota of wounds compared to the control and for the other types of wounds, indicating that these wounds are inhabited by fungal communities, often dominated by a few species, among which *T. atroviride* plays a major role (Figure 2).

**Table 5.** α-diversity indexes values calculated per every type of sample. Abbreviations: J—Suhora forest district, Jasionów forest area, K—Kudłoń forest district, Za Palacem forest area, P—Kudłoń forest district, Pasieka forest area, S—Suhora forest district, Czarny Groń forest area.

| Location | Type of Wound | Part of Wood | α-Diversity Indexes | | |
|---|---|---|---|---|---|
| | | | Shannon-Wiener (H) | Pielou (J) | Simpson (D) |
| J | old | outside | 0.0000 | | 0.0000 |
| J | fresh | inside | 0.4101 | 0.5917 | 0.2449 |
| J | fresh | outside | 0.6902 | 0.9957 | 0.4970 |
| J | fresh | middle | 0.4101 | 0.5917 | 0.2449 |
| K | control | inside | 0.9831 | 0.7091 | 0.5139 |
| K | control | outside | 0.7078 | 0.6443 | 0.4224 |
| K | control | middle | 1.0693 | 0.7713 | 0.5730 |
| K | old | inside | 1.6771 | 0.8618 | 0.7800 |
| K | old | outside | 1.8821 | 0.8566 | 0.8180 |
| K | old | middle | 1.7873 | 0.8595 | 0.7956 |
| K | healed | inside | 0.6229 | 0.4493 | 0.3033 |
| K | healed | outside | 0.4258 | 0.3876 | 0.2037 |
| K | healed | middle | 0.3488 | 0.5033 | 0.1975 |
| K | fresh | inside | 1.6408 | 0.9157 | 0.7846 |
| K | fresh | outside | 1.4594 | 0.8145 | 0.7160 |
| K | fresh | middle | 0.6931 | 1.0000 | 0.5000 |
| P | old | inside | 0.0000 | | 0.0000 |
| P | old | outside | 0.2062 | 0.2975 | 0.0997 |
| P | old | middle | 0.0000 | | 0.0000 |
| P | healed | inside | 2.2745 | 0.9153 | 0.8698 |
| P | healed | outside | 1.9073 | 0.9172 | 0.8194 |
| P | healed | middle | 1.2425 | 0.8962 | 0.6667 |
| P | fresh | inside | 0.0000 | | 0.0000 |
| P | fresh | outside | 1.0889 | 0.7855 | 0.5800 |
| S | control | inside | 1.2309 | 0.8879 | 0.6746 |
| S | control | outside | 0.5623 | 0.8113 | 0.3750 |
| S | control | middle | 0.8699 | 0.7918 | 0.5128 |
| S | old | inside | 1.6627 | 0.8545 | 0.7756 |
| S | old | outside | 1.7293 | 0.8316 | 0.7686 |
| S | old | middle | 1.8943 | 0.8621 | 0.8278 |
| S | healed | inside | 0.7356 | 0.6696 | 0.4063 |
| S | healed | outside | 0.6730 | 0.9710 | 0.4800 |
| S | healed | middle | 0.6365 | 0.9183 | 0.4444 |
| S | fresh | inside | 2.0577 | 0.9365 | 0.8549 |
| S | fresh | outside | 1.9568 | 0.8906 | 0.8375 |
| S | fresh | middle | 1.9437 | 0.9347 | 0.8443 |

**Table 6.** Sørensen's β-biodiversity index comparing the similarity of mycobiota inhabiting different types of bark-stripping wounds caused by red deer on silver fir in the Gorce National Park.

| Control | Fresh | Healed | Old |
|---|---|---|---|
| 0.43 | | | 0.38 |
| | | 0.42 | |
| | 0.34 | | |
| | | 0.63 | |
| | 0.59 | | |

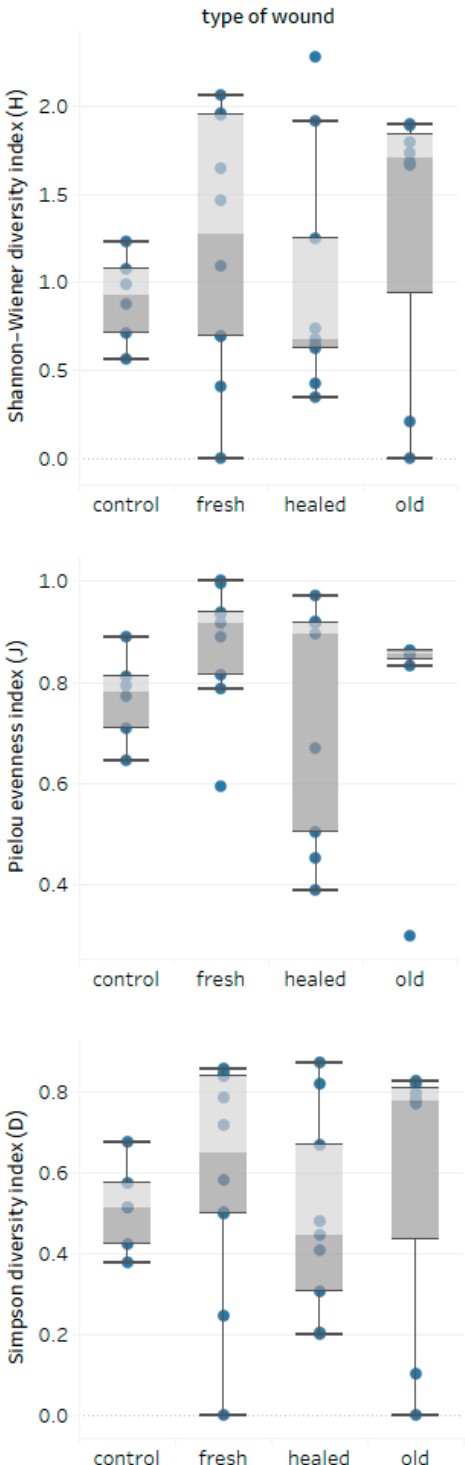

**Figure 7.** Box-plot representation of the biodiversity in three types of wounds of fir trunk, based on α-diversity indexes: Shannon-Wiener diversity index (H), Pielou evenness index (J) and Simpson diversity index (D).

On the basis of the Sørensen's β-biodiversity index, it was estimated that the most unique mycobiota were found in healed wounds that were least similar to controls (0.34) or old wounds (0.38) (Table 6). Fresh and old wounds showed the highest group similarity index (0.63).

## 4. Discussion

In the case of fresh wounds, species such as *A. alternata* and *A. arundinis* had the highest share. Fungi belonging to the genus *Phomopsis* as well as *S. polyspora* and *E. nigrum* were also isolated. These fungi, considered to be secondary pathogens, can also act as saprotrophs. They were also isolated by other researchers in wood at the initial stages of decomposition [39,40]. Particularly noteworthy is the large share of *A. arundinis* fungus in fresh wounds, the anti-fungal activity of which has been documented [41]. This fungus was also isolated in research on the mycobiota of fallen spruces [6]. Scar wounds were most frequently inhabited by *T. atroviride*, which was also dominant in the mycobiota of wood collected from control trees. As shown by recent molecular studies, this species may be a widespread endophyte [42,43], showing anti-fungal properties, which have been recently used to protect trees against wood-decaying fungi [44–47]. In Gorce, *T. atroviride* also numerously inhabited the wood of bark-stripped firs with a DBH above 13.0 cm. Perhaps the remarkable natural regenerative abilities of silver fir is the consequence of *T. atroviride* domination in the mycobiota of healed wounds of thicker trees and in the wood of visually undamaged control trees with DBH > 30.0 cm. As shown by the research on the dynamics of stands carried out in the Gorce National Park on permanent monitoring plots [14], despite the persistent strong pressure of red deer for years, expressed in the share of bark-stripped firs with DBH < 17.0 cm at the level of 80% of their numbers [15,48], in the last 25 years, the density of this forest-forming species in the tree layer has increased almost threefold, from 78 trees/ha in 1992 to 216 trees/ha in 2017, and the percentage share of fir in the stand increased from 16% to 36% [48]. The survival rate of this species, estimated on the basis of the number of trees reaching DBH 7.0–16.9 cm in 1992 that survived until 2017, ranges from 93% to 100%, depending on the thickness class (Table 7).

**Table 7.** Survivability of young silver fir trees in the area of the Gorce National Park in a period 1992–2017 assessed on a net of 433 permanent monitoring plots (0.05 ha each) distributed in the whole area of the park (for methodology see e.g., Chwistek 2010). Abbreviations: C—man-cut trees, BW—broken and windthrown trees.

| DBH [$d_{1,3}$; cm] in 1992 | Number of Silver Fir Trees in 1992 | Number of the Same Silver Fir Trees in 2017 | Mortality [N] | Survival Rate [%] |
|---|---|---|---|---|
| 7.0–7.9 | 164 | 153 | 11 (1C) | 93.3 |
| 8.0–8.9 | 116 | 110 | 6 (2C) | 94.8 |
| 9.0–9.9 | 113 | 110 | 3 (1C) | 97.3 |
| 10.0–10.9 | 71 | 69 | 2 (1C) | 97.2 |
| 11.0–11.9 | 71 | 70 | 1 | 98.6 |
| 12.0–12.9 | 82 | 80 | 2 (1BW) | 97.6 |
| 13.0–13.9 | 46 | 45 | 1 (1BW) | 97.8 |
| 14.0–14.9 | 40 | 39 | 1 | 97.5 |
| 15.0–15.9 | 44 | 44 | 0 | 100 |
| 16.0–16.9 | 38 | 38 | 0 | 100 |

In turn, Pach [29], while examining the influence of red deer on the cultivation values of fir stands in the Beskid Sądecki, another range of the Polish Carpathians, came to the assumption that the occurrence of damage as a result of bark-stripping in fir regeneration in the undergrowth stage, is probably not dangerous for trees, as compared to the damage in later development stages. In light of the above, the results of the studies by Niemtur et al. [13] who, while assessing the health of old fir trunks by using acoustic tomography method, found that in a random sample (mean age 99 years; N = 30) in the Gorce National Park, almost 80% of trees did not have butt rot. The share of healthy wood in this part of measured trunks reached 94%, despite the fact that the dimensions of the Gorce fir trees in the entire collection of 450 trees in the entire fir range in Poland were the highest.

The fungi of the genus *Eutypa* had a relatively large share in the mycobiota of fir trees that underwent bark-stripping. They dominated in the case of thinner trees and

were isolated from old wounds. They are dangerous pathogens for many crops that can also infect wood and occur together with other species of wood-decaying fungi [49–51]. The conducted research recorded the presence—along with *Eutypa* spp.—of the fungus *C. evolvens* in old wounds, which may participate in wood decomposition [52,53]. A similar situation occurs in the case of the *Phomopsis* genus, which very often colonize the wounds of fir [54,55] and are known carcinogenic fungi [56].

The actual role of inter-species competition of fungi in fir bark-stripped by red deer and the assessment of the importance of fungi showing inhibitory properties in the natural defense mechanism of this forest-forming species against destructive fungi will be the subject of a separate study.

## 5. Conclusions

- The greatest number of fungal colonies inhabiting the silver fir wood was obtained from old open wounds, with a smaller number of taxa compared to fresh and healed wounds.
- The age and DBH of bark-stripped trees do not seem to be of importance for the degree of infection of wounds expressed by the number of colonies; nevertheless, species diversity of wood decay fungi assessed by the α-diversity Shannon-Wiener (H) and Simpson (D) indices seems to be lower in healed wounds.
- The strong presence of *Arthrinium arundinis* and *Trichoderma atroviride* in firs wounds, which are often antagonistic to pathogenic fungi, may indicate the natural defense mechanisms of fir, aimed at inhibiting disease processes and, subsequently, the decomposition of wood.
- The high survival rate of the young generation of fir indicates that despite the strong pressure of red deer, the fir regeneration mechanism based on inter-species competition of fungi may be an effective tool in the fight for the survival of this species in the Carpathians.

**Author Contributions:** Conceptualization—W.P. and P.C.; methodology—W.P., A.B.-C., A.K.-P. and P.C.; investigation—W.P., A.B.-C., A.K.-P. and P.C.; formal analysis—K.P.; writing—original draft preparation—W.P., A.B.-C., A.K.-P., K.P. and P.C.; writing—review and editing—W.P., A.K.-P. and P.C.; supervision—W.P. All authors have read and agreed to the published version of the manuscript.

**Funding:** The research was funded by the Forest Fund under the agreement concluded between the State Forests National Forest Holding and the Gorce National Park in 2020 (agreement no. EZ.0290.1.6.2020, entitled "Activities of nature protection using forest management methods and other activities carried out in the Gorce National Park in 2020").

**Institutional Review Board Statement:** Not applicable.

**Informed Consent Statement:** Not applicable.

**Data Availability Statement:** Data sharing not applicable.

**Acknowledgments:** The authors would like to thank the Forest Fund for funding and financial support.

**Conflicts of Interest:** The authors declare that they have no conflict of interest.

**Ethics:** The samples in the study were collected on the area that belonged to the Gorce National Park and the authors received full permission to conduct the study. The experimental materials did not involve any humans or animals.

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
