# Peer review of "Is the Survivability of Silver Fir under Condition of Strong Ungulate Pressure Related to Mycobiota of Bark-Stripping Wounds?"

_forests, doi:10.3390/f12080976_

Round 1
Reviewer 1 Report
The manuscript is representing original research.
The authors discussing the state-of-the-art of silver fir tree growth in the Carpathians and the influence of deer on tree growth. The authors do not open the discussion and do not introduce the reader to understand the current situation with the diversity of fungi in natural ecological systems. At least a small introduction to the processes should be done. Another concern regarding the operation of the terms. For example, on page 2 line 68 authors use the term “decomposing” fungi when below the “decay” and “discolouration” fungi in line 76, 77. Page 2 line 78-83 are confusing and do not belong to the research questions which are purely described.
Page 3 line 103 the DBH index is confusing since authors do not describe whose breast height it is and where that index come from.
Page 4 line 123 probably related to the problem statement that should be in the introduction part. Moreover, the authors mentioned microscopic identification and DNA as a supplement to it but there are no microscopic analysis in the current manuscript discussed.
Page 16 line 260-283 more suitable to the introduction part in order to introduce readers to the stated problem.
Page 17 line 333-336 the manuscript does not show the comparison of classical morphological and DNA analysis therefore that statement is incorrect.
Page 17 line 344 is speculation since the antagonistic behaviour of those fungi against other fungi is not presented in the current study. The abundance of Trichoderma might be explained by the moisture conditions since that fungus grows a high water activity environment, mycoparasitism and the ability to decompose complex carbohydrates by opening access to successions of other wood-decaying fungi.
Reviewer 2 Report
The article „Fungi associated with the bark-stripping wounds of silver fir 2 and their potential significance for the survivability of this for-3 est species under conditions of strong ungulate pressure – the 4 case of the Gorce National Park (Western Carpathians, Poland)“ brings the basic overview of fungi species in bark stripped silver firs in Western Carpathians. The game damage is one of actual topics, however the article has few shortcomings.
Title of the article should characterize the article content shortly. I recommend to shorten this long title.
The introduction of the article should be focussed on the description of bark-stripping phenomenon. There should be mentioned which deer species cause this type of game damage. The introduction should also mention the impact of bark stripping on tree radial growth, rot development and growth in relation to climatic factors like precipitation and temperatures. The studies focused on silver fir are limited, however authors could also mention bark-stripping impact on Norway spruce.
The authors claim that the forest stands were under strong ungulate pressure. I miss the evidence about red deer population density in study area. Moreover, it is not clearly mentioned that the bark-stripping is caused by red deer (Cervus elaphus).
The article should contain clear hypotheses.
In the Material and methods have to be mentioned how the sampled trees were chosen.
One of major shortcomings are related to age of wounds which could be identified by dendrochronology (how the trees were sorted to three categories?). The soil type of stands is not mentioned and the phytocenology is also not stated. Moreover, the range of bark-stripping was not measured. The position of the tree (Predominant/dominant/subdominant…) was also not listed.
The article should be much more focussed on antagonistic to pathogenic fungi, however it could be one of hypotheses.
I have also some specific comments:
L 20 which deer species?
L 21 give the ammount of each mycobiota in percents
L 26 there should be not the assumptions in abstract (like "it seems that..."), only facts
L 28-29 You did not evaluate the radial growth of wood-decaying fungi in the trunks and impact on wood production or on the radial growth. Therefore, you can not clearly say if the deer pressure can or could not be the factor influencing the fir dynamics.
L 30 the keywords whould be not repeated from tittle of the article
L 41 The description and history of Gorce National Park should be mentioned in the Matherials and Methods section, not in the introduction
L 46-50 Give the citation for this statement
L 55 By red deer? The deer species must be clearly specified in the introduction with detail description of bark-stripping.
L 78 At the end of the Introduction should be clearly defined main hypotheses of the article point by point.
L 89 How was the categories of wounds sorted? It is evident, that the "fresh wound" on the picture 2 is few years old. The edge parts of the bark stripped wound in regeneration process and therefore the tree was bark stripped 2 or 3 years ago. It should be possible to identify the year when the tree was bark-stripped by wood samples taken by Pressler drill.
L 103 Have you measure the extent of the wound? The range of wound is one predictors for fungi colonization.
L 104 How were the sampled trees chosen? Have you evaluate also the stands? E.g. the trees which grow on the water-rich stands could inclinate to fungi colonization. On the other hand, the trees on dry stands could be much resistant to fungal pathogens.
L183 Table 2 should be listed as a supplementary file.
Round 2
Reviewer 2 Report
Dear authors,
thank you very much for improving your article. I believe that article is more understandable now. I have no further comments.